# The Impact of Low-Temperature Inactivation of Protease AprX from *Pseudomonas* on Its Proteolytic Capacity and Specificity: A Peptidomic Study

Chunyue Zhang [1,2,3], Sjef Boeren [4], Liming Zhao [2,3], Etske Bijl [1] and Kasper Hettinga [1,*]

[1] Dairy Science and Technology, Food Quality and Design Group, Wageningen University and Research, P.O. Box 17, 6700 AA Wageningen, The Netherlands
[2] State Key Laboratory of Bioreactor Engineering, School of Biotechnology, East China University of Science and Technology, Shanghai 200237, China
[3] Shanghai Collaborative Innovation Center for Biomanufacturing Technology (SCICBT), Shanghai 200237, China
[4] Laboratory of Biochemistry, Wageningen University and Research Center, Stippeneng 4, 6708 WE Wageningen, The Netherlands
[*] Correspondence: kasper.hettinga@wur.nl; Tel.: +31-317-482401

**Abstract:** The destabilization of UHT milk during its shelf life can be promoted by the residual proteolytic activity attributed to the protease AprX from *Pseudomonas*. To better understand the hydrolysis patterns of AprX, and to evaluate the feasibility of using low-temperature inactivation (LTI) for AprX, the release of peptides through AprX activity on milk proteins was examined using an LC-MS/MS-based peptidomic analysis. Milk samples were either directly incubated to be hydrolyzed by AprX, or preheated under LTI conditions (60 °C for 15 min) and then incubated. Peptides and parent proteins (the proteins from which the peptides originated) were identified and quantified. The peptides were mapped and the cleavage frequency of amino acids in the P1/P1′ positions was analyzed, after which the influence of LTI and the potential bitterness of the formed peptides were determined. Our results showed that a total of 2488 peptides were identified from 48 parent proteins, with the most abundant peptides originating from κ-casein and β-casein. AprX may also non-specifically hydrolyze other proteins in milk. Except for decreasing the bitterness potential in skim UHT milk, LTI did not significantly reduce the AprX-induced hydrolysis of milk proteins. Therefore, the inactivation of AprX by LTI may not be feasible in UHT milk production.

**Keywords:** UHT milk; shelf life; *Pseudomonas*; AprX protease; enzyme inactivation; peptidomics

## 1. Introduction

The storage of raw milk under refrigeration conditions, for example, in a milk tank on a farm, allows the growth of psychrotrophic bacteria [1]. Among all psychrotrophic bacteria that may grow, the *Pseudomonas* genus is predominant in raw milk [2–4]. The majority of pseudomonads can produce a specific extracellular metalloprotease, AprX, which can hydrolyze α$_s$-, β-, and κ-caseins [5]. In addition, AprX is highly heat resistant and can therefore remain active after ultra-high-temperature (UHT) processing (typically 135–150 °C for 2–10 s) [6,7]. As a result, different types of UHT milk destabilization, such as age gelation, fat separation, and a bitter flavor, have been attributed to AprX activity [8–11].

In order to decrease this instability of UHT milk during its shelf life, the AprX level should be minimized by implementing good hygiene practices to control the pseudomonads that can produce AprX and by enhancing the inactivation of AprX by adjusting UHT processing, especially in milk destined for export or stored at high ambient temperatures. Currently, there are two main mechanisms that can play a role in the inactivation of AprX. The first is thermal inactivation, which, for the highly heat resistant AprX, refers to the inactivation at UHT temperatures. However, the D-value (the time required to reduce

the enzyme activity to 10% of its original value) for the inactivation of AprX is as high as 2.7 ± 1.4 min at 130 °C [6], meaning that reducing AprX activity using current UHT regimes (2.4 s at 138 °C) is difficult without introducing the detrimental effects of increased heating to its sensory properties and to the heat-sensitive whey proteins. The second mechanism is low-temperature inactivation (LTI), which is based on the fact that the autolysis of AprX can occur at lower temperatures corresponding roughly to the denaturation temperature (Td). At such temperatures, AprX enzyme molecules are present as a mixture of (1) folded, active, proteolytic, compact molecules and (2) (partially) unfolded, inactive molecules. The native state can exempt the enzymes from autolysis, while the (partially) unfolded structure renders the enzyme susceptible to intermolecular autolysis, thereby decreasing the AprX activity [6,12].

LTI treatments at 50–60 °C for inactivating bacterial proteases, especially AprX, have been reported in many studies [12–18]. However, conflicting results have been obtained in these studies with respect to the effects of LTI on extending the shelf life of UHT milk. Driessen [19] reported that preheating the milk for 60 min at 55 °C before UHT processing could improve the quality of UHT milk by retarding the gelation, proteolysis, bitterness, and transparency of the milk. Bitterness due to the formation of hydrophobic peptides is frequently encountered in UHT milk [20,21]. In contrast, Kocak and Zadow [22,23] found that LTI before UHT processing did not significantly inhibit proteolysis during the storage of UHT milk, but LTI after UHT treatment doubled the shelf life by retarding the onset of age gelation. A three-fold extension of the shelf life was also reported by West et al. [15], who LTI-treated skim milk after UHT treatment.

The inconsistency observed in the inhibition of proteolysis in LTI-treated UHT milk samples may have been caused by the physicochemical changes being partly dependent on the initial proteolysis of the milk protein system [22] because, during LTI, besides AprX itself, the milk proteins can also be hydrolyzed as substrates. An LTI-based autolysis can only be a useful step in the production process of UHT milk if the hydrolysis of the milk proteins during LTI is less than the hydrolysis that would have been caused by the inactivated AprX. Otherwise, the LTI treatment may effectively increase the hydrolysis of milk proteins compared to an untreated product, which would in turn lead to faster protein hydrolysis, finally resulting in an earlier development of destabilization during storage [6]. However, the hydrolysis of milk proteins during LTI and subsequent incubation remains poorly studied. Studying the hydrolysis of milk proteins in detail may be achieved by using peptidomic technology. This technology has previously been used to examine the quality of dairy products by identifying and quantifying peptides in milk [24], e.g., to examine the proteolytic activity of microorganisms on bovine milk proteins [25]. A recent investigation using peptidomics to study the AprX-induced destabilization of UHT milk showed no variation in the peptides formed by AprX when studying different *Pseudomonas* strains [26]. This study, as well as another peptidomic study on AprX, found peptides from different caseins to be formed preferentially [26,27]. Hence, a peptidomic approach was employed in the current study to further elucidate the milk protein hydrolysis pattern based on the peptides formed.

In addition, a mechanistic understanding of the casein hydrolysis patterns by AprX is still limited. Earlier studies often assessed the hydrolysis of milk proteins by AprX qualitatively, e.g., by counting the numbers of identified peptides. The order of cleavage sites for each casein was found to be $\beta$- > $\alpha_{s1}$- > $\kappa$- > $\alpha_{s2}$-caseins [9,11,26,28], whereas studies that characterized the decrease in intact caseins showed the order to be $\kappa$- > $\alpha_s$- > $\beta$-caseins using electrophoresis [29,30], as well as $\kappa$- > $\beta$- > $\alpha_s$-caseins using RP-HPLC [5] and electrophoresis [30,31]. Given the inconsistent conclusions drawn from these different analytical approaches, there is a need for a more systematic, quantitative analysis using up-to-date mass-spectrometric detection methods to better discriminate the preference of AprX towards specific parts of the individual casein molecules.

Therefore, this study first aimed to increase the knowledge of AprX hydrolysis patterns on milk proteins by using a quantitative peptidomic perspective. The second aim was to

determine the feasibility of using an LTI treatment as a means of inactivating AprX in both full-fat and skim UHT milk by comparing the peptidome of AprX-hydrolyzed UHT milk samples with and without an LTI treatment.

## 2. Materials and Methods

### 2.1. Materials

All chemicals were of analytical grade and purchased from Sigma-Aldrich (St. Louis, MO, USA) or Merck (Darmstadt, Germany).

### 2.2. Determination of the LTI Condition

AprX was isolated from the strain *Pseudomonas fluorescens* Migula 1895 (DSM 50120, Deutsche Sammlung von Mikroorganismen, Braunschweig) using the method described by Zhang et al. [5]. An amount of 2 mg of crude AprX was heated for 15 min at different temperatures ranging from 40 °C to 80 °C in 10 mL of skim or full-fat UHT milk, as well as 10 mM phosphate-buffered saline (PBS, pH of 6.7). The holding time was 15 min because the extent of the inactivation of AprX, reflected by the decrease in proteolytic activity, did not markedly increase when heated for longer than 15 min (based on the results of pre-experiments; data not shown). Samples were immediately cooled on ice for 10 min to stop the LTI treatment. A control sample prepared without LTI heating was taken as a reference, corresponding to 100% proteolytic activity. The proteolytic activity was determined at 42 °C, which is the optimal temperature for AprX, with the azocasein assay as described before [5]. Assays were performed in triplicate.

### 2.3. Sample Preparation

Commercial skim and full-fat UHT milk was purchased from a local supermarket. These milk samples had been subjected to direct sterilization by a steam-infusion heat treatment. To keep a low starting degree of hydrolysis, we used milk samples that were not older than 1 month after manufacture. To inhibit the growth of microorganisms, 0.02% sodium azide and 0.0005% bronopol were added. The milk composition was determined to be 3.76 and 3.77 g of protein/100 mL milk, and 0.07 and 3.67 g of fat/100 mL milk in skim and full-fat UHT milk, respectively, as determined by MilkoScan 134A/B (Foss Electric, Hillerød, Denmark). Skim and full-fat UHT milk samples without the addition of AprX were used as untreated controls (coded SM for skim milk and FM for full-fat milk). Crude AprX extract was added to the other samples at a concentration of 0.2 mg/mL of milk. The samples without LTI treatment were coded SM + AprX and FM + AprX. The samples with AprX that were first treated with LTI and then incubated were coded SM + AprX + LTI and FM + AprX + LTI. All samples were prepared in triplicate and incubated at 42 °C for 5 h in order to allow the milk proteins to be hydrolyzed by the untreated and LTI-treated AprX.

### 2.4. Peptide Extraction

Peptides were extracted using the method previously described [32], with modifications. Briefly, right after the incubation, the samples were mixed with the same volume of 200 g/L trichloroacetic acid (TCA) solution. The samples were mixed by vortex at high speed for 10 s, then centrifuged at 3000× *g* at 4 °C for 10 min, after which both intact proteins and fat were removed by the TCA precipitation. The supernatant was collected and cleaned by a solid phase extraction (SPE) C18 column clean-up step prior to the analysis by LC-MS/MS. Stage tips containing LiChroprep C18 column material LiChroprep® RP 18 (25–40 μm) (Merck KGaA, Darmstadt, Germany) were made in-house as described previously [33]. The C18+ stage tip column was washed 2 times with 200 μL of methanol, and then 4 μL of 50% column material LiChroprep C18 in methanol was added. The μColumn was conditioned with 100 μL of 1 mL/L formic acid (HCOOH). Then, 50 μL of the peptide samples were loaded on the C18+ stage tip column followed by washing with 100 μL of 1 mL/L HCOOH. The peptides were eluted with 50 μL of 70% acetonitrile/30% 1 mL/L HCOOH from the C18+ stage tip column. The samples were then dried in a vacuum

concentrator (Eppendorf Vacufuge®, Eppendorf, Hamburg, Germany) at 45 °C for 20 to 30 min until the volume of each sample decreased to 15 µL or less. The samples were reconstituted to 50 µL with 1 mL/L HCOOH for analysis by LC-MS/MS. Validation of the reproducibility of the peptide extraction method has been previously reported before [32].

### 2.5. LC-MS/MS-Based Peptidomic Analysis

An analysis of the peptide extracts was performed on a nano-LC/LTQ-OrbitrapXL (Thermo Fisher Scientific, Bremen, Germany) system. A sample volume of 18 µL was injected onto a pre-concentration column (prepared in-house) and the peptides were eluted onto a 0.10 × 200 mm Magic C18 resin analytical column with an acetonitrile gradient at a flow rate of 0.5 µL/min. The gradient elution increased from 5% to 30% acetonitrile in water with 1 mL/L HCOOH in 50 min. The column was then washed using a fast increase in the percentage of acetonitrile to 50% (with 50% water and 1 mL/L HCOOH in both the acetonitrile and the water) in 3 min. A P777 Upchurch micro-cross was positioned between the pre-concentration and analytical column. An electrospray potential of 3.5 kV was applied directly to the eluent via a stainless-steel needle fitted into the waste line of the micro-cross. Full-scan positive-mode FTMS spectra were obtained in the LTQ-Orbitrap XL (Thermo electron, San Jose, CA, USA) between an m/z of 280 and 1400 at a resolution of 15,000. MS/MS scans of the most abundant singly, doubly, and triply charged peaks in the FTMS scan were recorded in the data-dependent mode in the Orbitrap at a resolution of 7500 (MS/MS threshold = 10,000, 45 s exclusion duration) using a contaminant m/z mass list to prevent the selection of contaminants.

### 2.6. Peptide and Protein Identification and Quantification

The MS/MS spectra from each run were analyzed using MaxQuant v.1.6.3.4 with the Andromeda search engine [34]. Peptides and parent proteins (the proteins from which the peptides originated) were quantified based on label-free quantification (LFQ) with a minimum ratio count of 2 peptides, of which at least one should be unique for proteins. The match between runs and unspecific digestion settings were used. A false discovery rate (FDR) of 0.01 was set for both the peptide spectrum match level and the protein level. A previously established bovine serum protein database [35] was used for the Andromeda searches with a minimum peptide length of 7 and a maximum length of 25 amino acids. A standard contaminants list containing human keratin and trypsin sequences was included in the search. The variable modifications used included the phosphorylation of serine, threonine, and tyrosine; the oxidation of methionine; the acetylation of the protein N-terminus; and the deamidation of asparagine and glutamine.

### 2.7. Data Analysis

Filtering and statistical analyses on the MaxQuant output (Section 2.6) were performed with Perseus v.1.6.0.7 [36]. Both the proteinGroups.txt and peptides.txt output from MaxQuant were used, containing data on 18 measurements of skim milk and whole milk, including the samples without AprX, with Aprx, and with AprX + LTI, with all being analyzed in technical triplicate. Using Perseus, the data were filtered for the removal of reverse and non-bovine contaminant sequences, log10-transformed, and grouped by the fat content of the UHT milk, the addition of AprX, and the LTI treatment. Next, the data were filtered by rows based on valid values for a minimum of 50% occurrences of each protein or peptide sequence. Missing intensity values were then imputed across the entire matrix using random values from a normal distribution with a variation of 0.3 and a downshift of 1.8, which was meant to simulate expression below the limit of detection [36]. Once imputed, the values were normalized based on the z-score, which was calculated in Perseus by subtracting the median intensity from individual intensities, followed by division by the standard deviation. Differences in protein and peptide profiles were tested across AprX-hydrolyzed (with or without LTI treatment) groups by ANOVA, also using Perseus. For ANOVA, S0 was set to 0.1, with permutation-based FDR, where FDR was set

to 0.05, which are the default settings in Perseus. For volcano plots, two-sided *t*-tests were used to determine the differences caused by the LTI treatment. For this analysis, the FDR was set to 0.05 and S0 was set to 0.1. The S0 value here allowed for the testing of artificial variance within groups and controlled for differences between means, which required a larger absolute difference between groups [37]. Throughout, the results are reported as means ± standard deviation (SD) unless noted otherwise.

### 2.8. Enzyme Predictions

The enzymes responsible for the cleavage of proteins in blank UHT milk samples were predicted with the web-based software EnzymePredictor [38]. Enzymes were evaluated and classified based on the total number of cleavages and their odds ratio (OR), with the OR calculated as each enzyme's tendency to cut at termini rather than the interior of peptides. These values were used to determine the degree of participation of certain enzymes in the hydrolysis of the proteins identified. The N-terminal and C-terminal cleavages of individual parent proteins were also assessed.

### 2.9. Frequency of Amino Acids in the P1 and P1′ Position

The enzyme specificity of AprX was represented using the subsite nomenclature from Schechter and Berger [39]. Amino acid residues were designated as P1, P2, P3, P4, etc., in the N-terminal direction from the cleavage bond. Likewise, the amino acid residues in the C-terminal position were designated as P1′, P2′, P3′, P4′, etc., as shown in the following model:

$$\text{PN- - - - -P4-P3-P2-P1} \neq \text{P1′-P2′-P3′-P4′- - - - -PC′}$$

The scissile peptide bonds (the bond susceptible to cleavage) at the amino and carboxyl termini for each peptide were identified from the MS/MS analysis and the milk protein amino acid sequence. The frequency of amino acids in the P1 and P1′ position was studied based on all the peptides identified originating from five milk proteins ($\alpha_{s1}$-, $\alpha_{s2}$-, β-, and κ-caseins and β-lactoglobulin) in all the sample groups.

The cleavage frequency of every amino acid in the P1 or P1′ position was calculated using the number of detected peptide bonds involving a specific amino acid in the P1 or P1′ position, divided by the total number of peptide bonds involving this amino acid residue in the P1 or P1′ position [11].

### 2.10. Bitterness Predictions

The potential bitterness of peptides was evaluated by applying Ney's Q-rule based on peptide hydrophobicity [40]. An average hydrophobicity, Q, of the peptide was calculated as the sum of the amino acid side chains' hydrophobicity divided by the number of amino acid residues of the peptide. Peptides with a Q value > 1400 cal/mol were considered to be potentially bitter [40].

## 3. Results

### 3.1. Determination of Appropriate LTI Conditions

Before applying LTI to UHT milk, first the optimal conditions for LTI were determined. As shown in Figure 1, the activity of AprX decreased to a different extent after 15 min of pre-incubation at different temperatures from 40 to 80 °C. The most pronounced decrease in activity was found at 60 °C in skim and full-fat UHT milk and at 58 °C in PBS buffer, with 73%, 77%, and 57% residual activity, respectively. Heating at 60 °C for 15 min was thus used as the LTI conditions for our further experiments. AprX was inactivated to a larger extent in the PBS buffer than in the milk samples.

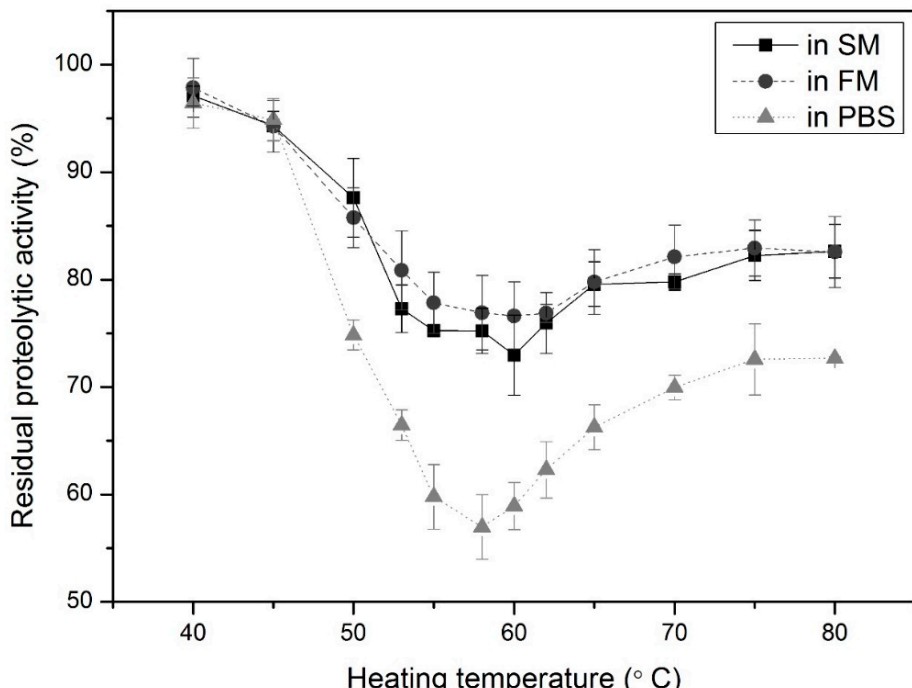

**Figure 1.** Residual proteolytic activity of AprX after thermal treatment in skim UHT milk, full-fat UHT milk, and 10 mM PBS (pH 6.7). Ten mL of UHT milk or PBS buffer with an addition of 2 mg of AprX was pre-incubated for 15 min at different temperatures from 40 to 80 °C. Results are expressed as the percentage of the enzyme activity without being heated. UHT: ultra-high temperature; SM: skim milk; FM: full-fat milk; PBS: phosphate-buffered saline.

*3.2. Proteins Being Hydrolyzed*

As reflected by the lower inactivation in milk than in PBS, during the LTI, milk proteins may also be hydrolyzed by native AprX. To study the impact of LTI on UHT milk proteins, proteins and peptides were profiled across the six sample groups. Parent protein profiles were qualitatively and quantitatively assessed. A total of 48 detected parent proteins were hydrolyzed to a total of 2488 identified peptides, as detailed in Table S1. The order of the main milk proteins according to the number of peptides formed was β-casein (738) > $\alpha_{s1}$-casein (576) > $\alpha_{s2}$-casein (279) > β-lactoglobulin (216) > κ-casein (190), with these five proteins representing approx. 80% of the total number of peptides. However, for quantitative comparisons, peptide LFQ intensity, rather than peptide presence, was used for further data analysis in this study.

To assess differences in parent protein levels between groups, hierarchical k-means clustering was performed. As shown in Figure 2a, large differences could be seen between the LFQ intensities of control UHT milk and milk incubated with AprX. Samples also clustered according to the fat content, but not according to the LTI treatment.

Comparing the total summed peptide LFQ intensity of the five most abundant proteins (Figure 2b), all the hydrolyzed samples were significantly higher ($p < 0.005$) than the untreated UHT samples. In addition, the control FM was found to be significantly lower than the control SM ($p < 0.01$), which may be due to different industrial processing methods of the different types of milk. Except for these, no significant differences in the total peptide intensities were found among the hydrolyzed groups, regardless of the fat content or LTI treatment, indicating that LTI did not significantly decrease the overall degree of hydrolysis.

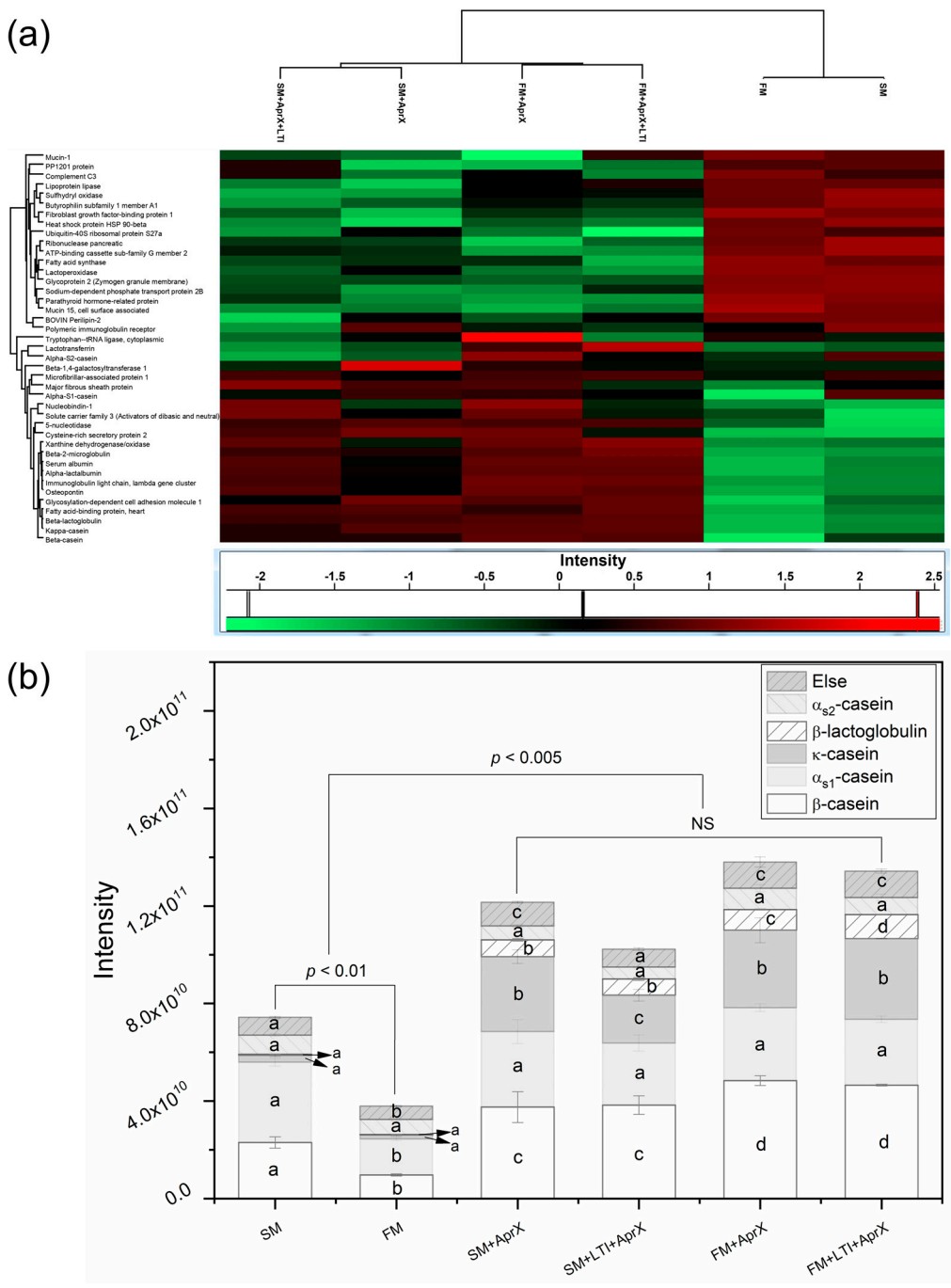

**Figure 2.** Quantitative hierarchical k-means clustering shows samples as the column tree and parent proteins as the row tree, with lower intensities plotted as green with an increasing color scale from green to red as the z-score normalized and the protein's LFQ intensity increased (**a**). Summed peptide LFQ intensity per parent proteins; different letters stand for significant differences in the summed peptide LFQ intensity of the individual protein between groups, with $p < 0.01$ (**b**). UHT: ultra-high temperature; SM: skim milk; FM: full-fat milk; LTI: low-temperature inactivation. SM + AprX and FM + AprX were added with crude AprX extract at a concentration of 0.2 mg/mL milk; SM + AprX + LTI and FM + AprX + LTI were treated with LTI (60 °C for 15 min) before incubation. All samples were prepared in triplicate and incubated at 42 °C for 5 h in order to allow the milk proteins to be hydrolyzed by untreated and LTI-treated AprX.

The summed peptide intensity of the five most abundant parent proteins ($\alpha_{s1}$-casein, $\alpha_{s2}$-casein, $\beta$-casein, $\kappa$-casein, and $\beta$-lactoglobulin) were also compared separately per protein

(Figure 2b). In control samples (SM and FM), the peptide intensity originated from the parent proteins in the order of $\alpha_{s1}$-casein > β-casein > $\alpha_{s2}$-casein > κ-casein > β-lactoglobulin. After the hydrolysis by AprX, the increase in peptide intensity was compared. It turned out that the total intensity of the peptides from κ-casein and β-casein increased by the highest amounts, in both skim and full-fat UHT milk. The intensity of the peptides from β-lactoglobulin also rose significantly ($p < 0.001$). There were barely any peptides detected in the control samples (Figure 2b), indicating that β-lactoglobulin mostly stayed intact in normal UHT milk, but it was hydrolyzed intensively by AprX.

On top of these five above-mentioned major milk proteins, peptides from 25 parent proteins in total were found to be significantly higher in intensity after the hydrolysis by AprX, as listed in Table S1. In these proteins, we found several milk fat globule membrane (MFGM) proteins, such as mucins and butyrophilin, and many enzymes, such as lactoperoxidase, lipoprotein lipase, fatty acid synthase, and sulfhydryl oxidase. This indicates that AprX can hydrolyze almost all proteins and is not specifically active on a limited number of proteins.

It is noteworthy that many peptides were already naturally present in the non-incubated control samples. The potential enzymes responsible for cleaving the proteins were predicted, showing that plasmin and trypsin were found to be the two enzymes with the highest likelihood of activity (Figure S1).

### 3.3. Peptides Formed by AprX

### 3.3.1. Peptide Mapping

After comparing the total peptide intensities of the five most abundant parent proteins in Section 3.2, we mapped the peptides of β-, κ-, $\alpha_{s1}$-, and $\alpha_{s2}$-caseins and β-lactoglobulin in relation to the overall protein sequence to visualize the hydrolysis patterns of AprX on these proteins. Other parent proteins were not further studied because they have no known influence on UHT milk stability. The peptide profiles of the FM and FM + AprX samples are shown in Figure 3 as examples of control and treated samples, respectively. The peptide profile of other control and treated sample pairs were similar to that in Figure 3, as shown in Figure S2. Regarding the control sample, FM, overall, a high-sequence coverage (β-casein, 100%; $\alpha_{s1}$-casein, 89%; $\alpha_{s2}$-casein, 89%; κ-casein, 67%; and β-lactoglobulin, 61%) was found. Most peptides that were detected originated from β- and $\alpha_{s1}$-caseins (Figure 3(A1,C1)), while fewer peptides came from $\alpha_{s2}$- and κ-casein (Figure 3(B1,D1)) and β-lactoglobulin (Figure 3(E1)). With regard to the hydrolyzed FM samples, β-casein was readily cleaved all over the molecule (Figure 3(A2)). A strong proteolysis also characterized κ-casein (Figure 3(B2)). Even though the number of identified peptides coming from κ-casein was much lower than the other caseins (Table S1), the total intensity of the peptides from κ-casein ranked second after β-casein (Figure 2b). Figure 3C,D show that the sequences of $\alpha_{s1}$- and $\alpha_{s2}$-caseins were almost entirely covered by the peptides detected, except for the phosphorylated regions.

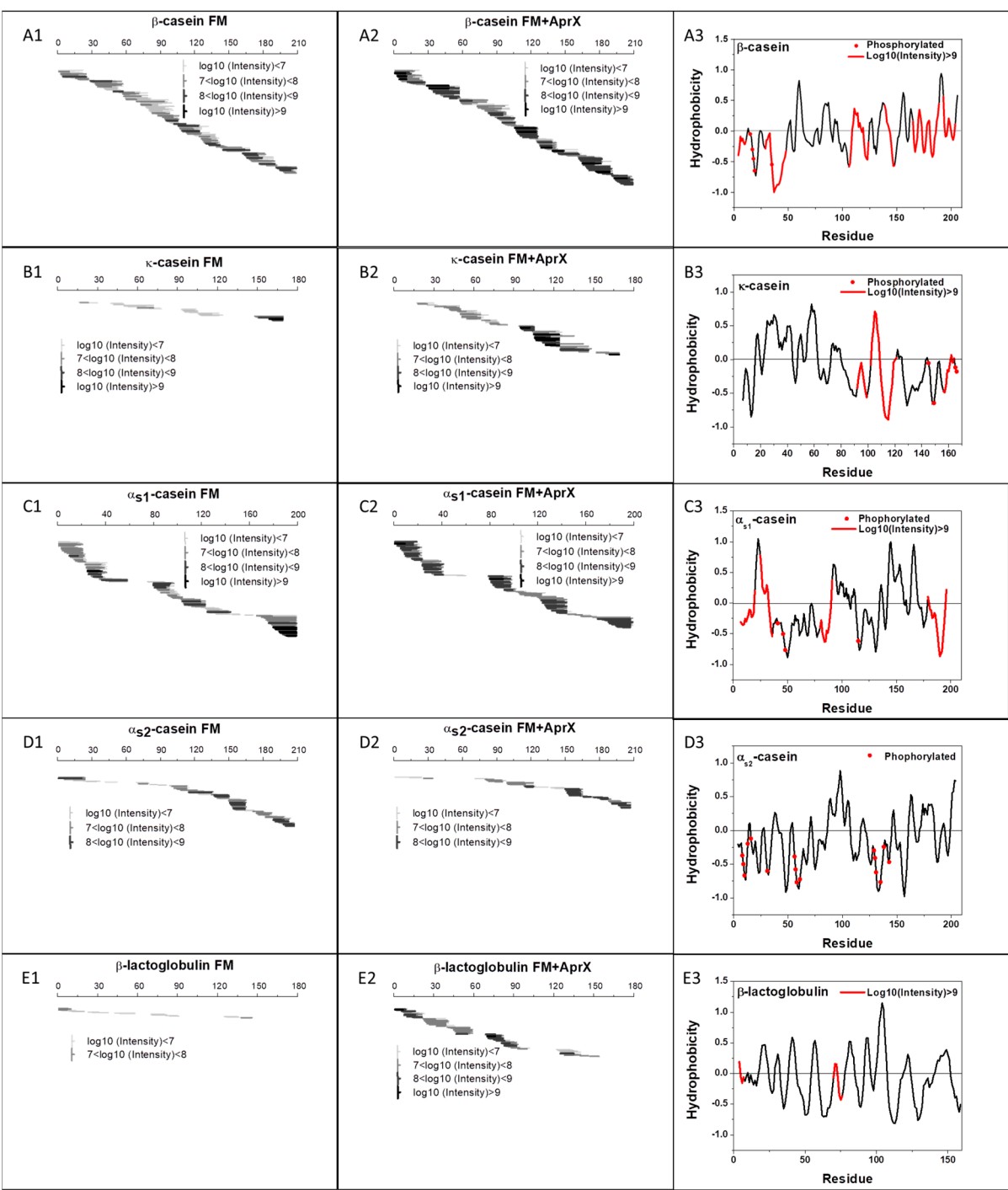

**Figure 3.** Peptidomic profiles of FM (1) and FM + AprX (2) and hydrophobicity distribution (3) of β-casein (**A**), κ-casein (**B**), $\alpha_{s1}$-casein (**C**), $\alpha_{s2}$-casein (**D**), and β-lactoglobulin (**E**), respectively. The peptidomic profiles were analyzed by a nano-LC/LTQ and Orbitrap. For the MS/MS, peptides with different intensities were distinguished by color and width, and the intensity shown in the figure is the average value of the results in triplicate. The hydrophobicity was calculated according to Sweet and Eisenberg (1983), where positive values indicate hydrophobic regions and negative values indicate hydrophilic regions. Peptides with an intensity higher than $10^9$ in FM + AprX are highlighted in red, and the detected phosphorylated clusters by LC-MS/MS are marked with (●). FM + AprX: full-fat milk with crude AprX extract added at a concentration of 0.2 mg/mL milk.

### 3.3.2. Cleavage at P1 and P1′ Positions

Figure 4 shows the cleavage frequency of amino acids in the P1 and P1′ position for all the peptides obtained after the hydrolysis of $\alpha_{s1}$-, $\alpha_{s2}$-, $\beta$-, and $\kappa$-caseins and $\beta$-lactoglobulin by AprX. AprX has a broad specificity, as made apparent from the peptide-mapping results (Figure 3). Overall, all the basic amino acid residues (His, Lys, and Arg), the aliphatic amino acid Leu, the sulfur-containing residue Met, and the aromatic amino acid Phe in both the P1 and P1′ positions led to a strong cleavage of the peptide bond. Conversely, the presence of acidic amino acid residues (Asp and Glu) or the sulfur-containing residue Cys in both the P1 and P1′ position, Ile in the P1 position, and Pro in the P1′ position appeared unfavorable for the cleavage of the peptide bond by AprX.

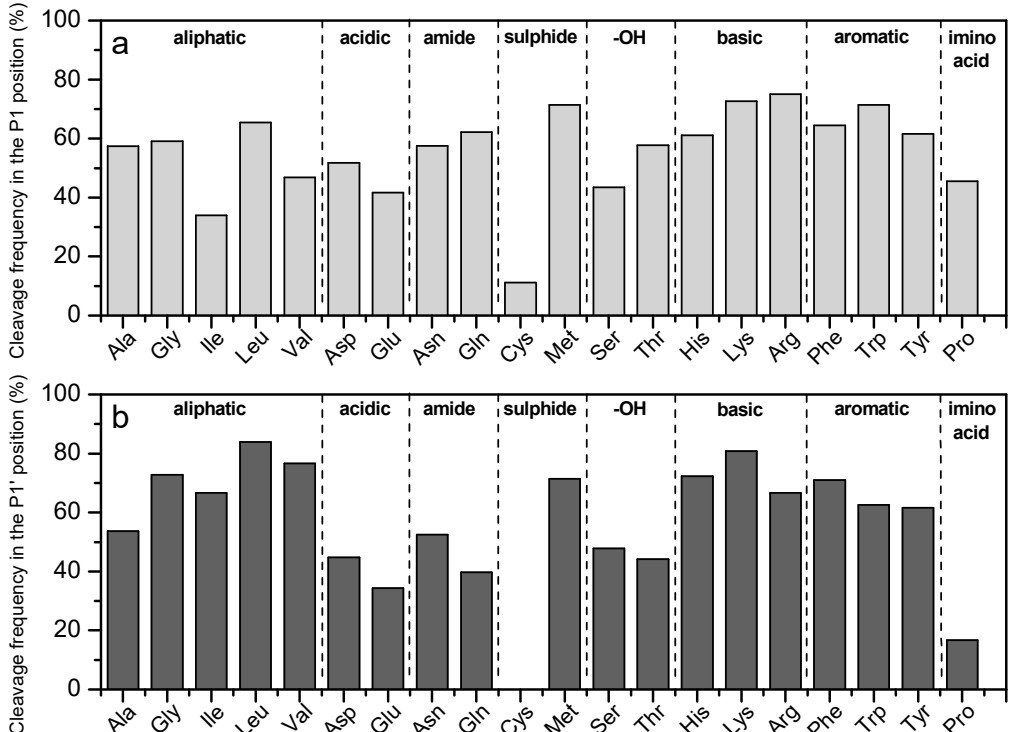

**Figure 4.** Cleavage frequency for each amino acid in the P1 (**a**) and P1′ (**b**) position according to their respective proportion. The analysis was based on the peptides identified from $\alpha_{s1}$-, $\alpha_{s2}$-, $\beta$-, and $\kappa$-caseins and $\beta$-lactoglobulin in all the AprX-hydrolyzed samples. Amino acids were grouped by a dashed line based on the side chain structure.

### 3.4. Effects of LTI

### 3.4.1. Comparison of Peptide Profiles

Even though the peptide mapping (Figure 3) showed similar patterns in AprX-hydrolyzed skim and full-fat milk, irrespective of LTI treatment, some peptide sequences were found to be significantly different by ANOVA when comparing samples with and without LTI treatment. Volcano plots (Figure 5) were made to visualize the differences of the peptide intensities between groups with and without LTI treatment; details of the significantly different sequences are presented in Table S2.

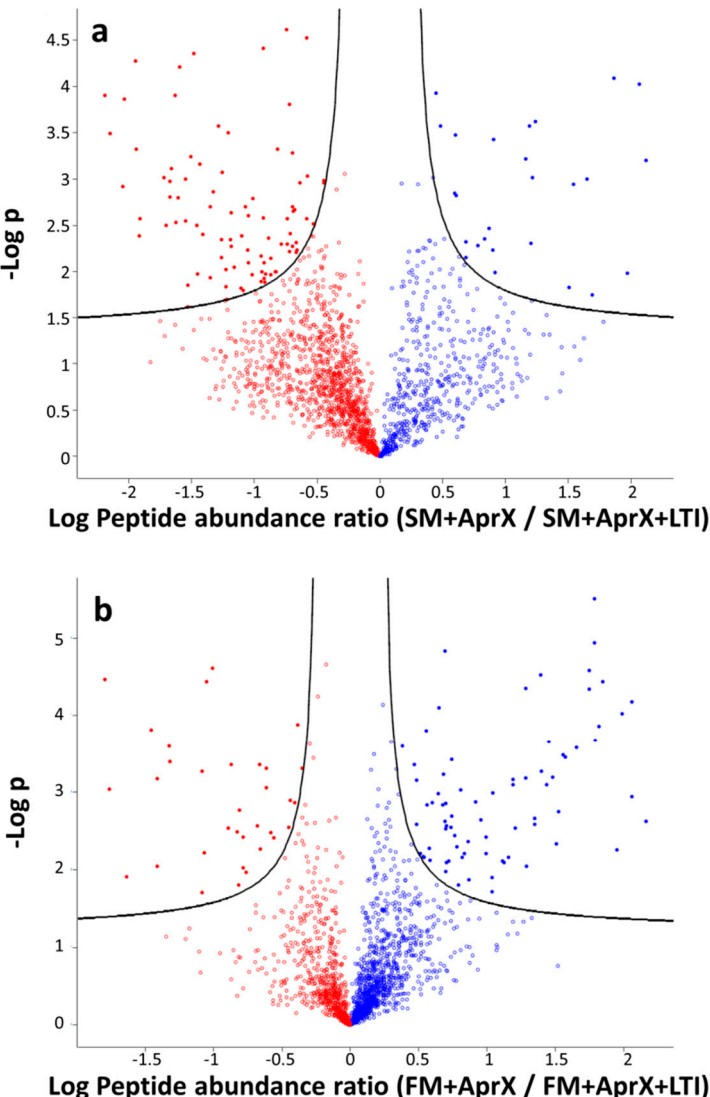

**Figure 5.** Volcano plots depicting log10-fold change in peptide intensity between groups (*x*-axis) and −log *p* value (*y*-axis). All significant peptides are represented as filled circles above the line, and non-significant peptides are unfilled below the line. Peptide intensity distribution between SM + AprX and SM + AprX + LTI; peptides higher in SM + AprX are red and peptides higher in SM + AprX + LTI are blue (**a**). Peptide intensity distribution between FM + AprX and FM + AprX + LTI; peptides higher in FM + AprX are red and peptides higher in FM + AprX + LTI are blue (**b**). SM: skim milk; FM: full-fat milk; LTI: low-temperature inactivation. SM + AprX and FM + AprX were added with crude AprX extract at a concentration of 0.2 mg/mL milk; SM + AprX + LTI and FM + AprX + LTI were treated with LTI (60 °C for 15 min) before incubation. All samples were prepared in triplicate and incubated at 42 °C for 5 h in order to allow the milk proteins to be hydrolyzed by untreated and LTI-treated AprX.

As shown in Table S2, in total, 115 and 108 peptides were found to be significantly different (*p* < 0.05) after LTI treatment in skim milk and full-fat milk, respectively. In the SM + AprX group, 89 peptides were found to be higher in intensity than in the SM + AprX + LTI group, while 26 peptides were higher in the LTI groups. In full-fat milk, however, the number of peptides with significantly higher intensities was found to be higher in the LTI group (75 peptides) than in the non-LTI group (33 peptides). Among all the different peptides, we paid particular attention to the caseinomacropeptide (CMP) fragments (sequences in the f (106–169) region in κ-casein), because these are very relevant for casein micelle stability. Based on our results (bold in Table S2), four (three higher in

SM + AprX and one higher in SM + AprX + LTI) and six (two higher in FM + AprX and four higher in FM + AprX + LTI) CMP fragments were found to be significantly different after LTI treatment in SM and FM, respectively.

### 3.4.2. Bitterness Prediction

In our study, a total of 877 peptides were identified to have a Q value above 1400 cal/mol, which is an evaluation criterion for a bitter peptide [21]. We multiplied the Q value of these potentially bitter peptides (Q > 1400 cal/mol) with their respective LC-MS/MS intensities in the different samples to quantitatively compare the bitterness potential. As shown in Figure 6, the bitterness potential of the groups hydrolyzed by AprX was significantly higher than that of the two blank groups because hydrolysis gave rise to the formation of many hydrophobic peptides, which can contribute to bitterness [41]. Comparing among the AprX-hydrolyzed groups, the SM + AprX group was found to have a higher bitterness potential than SM + AprX + LTI, indicating that the LTI treatment may reduce the formation of bitter peptides in skim milk, although the absolute differences between the groups were small. In full-fat milk, however, the LTI treatment did not influence the bitterness potential significantly.

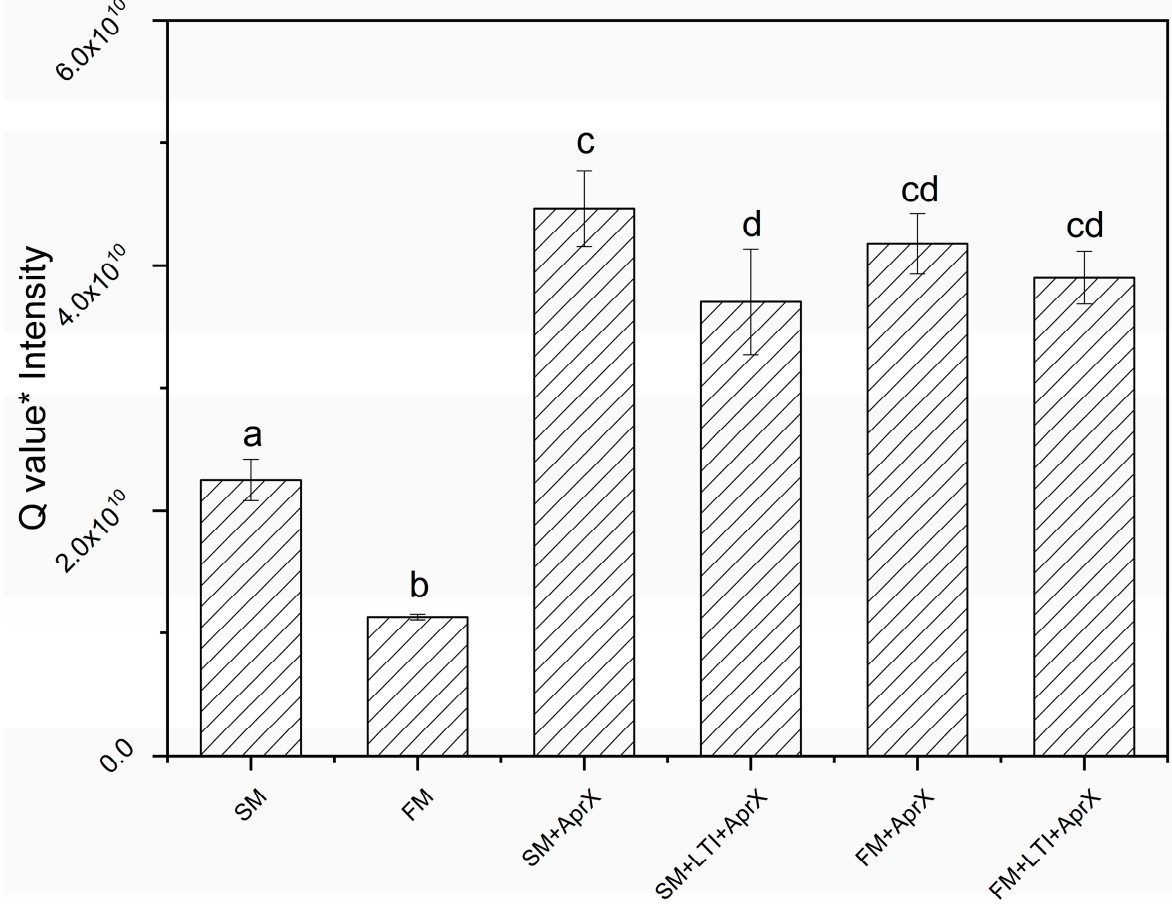

**Figure 6.** Quantitative comparison of bitterness potential, expressed as the sum of Q value * intensity of all the peptides with a Q value > 1400 cal/mol in the different samples, different letters stand for significant differences in the Q value * intensity between groups, with $p < 0.01$. SM: skim milk; FM: full-fat milk; LTI: low-temperature inactivation. SM + AprX and FM + AprX received an addition of crude AprX extract at a concentration of 0.2 mg/mL milk; SM + AprX + LTI and FM + AprX + LTI were treated with LTI (60 °C for 15 min) before incubation. All samples were prepared in triplicate and incubated at 42 °C for 5 h in order to allow the milk proteins to be hydrolyzed by untreated and LTI-treated AprX.

## 4. Discussion

Using a peptidomic approach, the detailed hydrolysis of milk proteins by AprX as well as the impact of LTI on that hydrolysis was studied.

Before collecting LTI-treated samples for peptidomic analyses, the optimal temperature for performing the LTI treatment was determined, which turned out to be 60 °C for both skim and full-fat milk (Figure 1). The decrease in AprX activity at 60 °C was most probably caused by the intermolecular autoproteolysis of the unfolded AprX molecules by native AprX molecules [12], as explained in the introduction, although this could not be confirmed based on our data. A similar temperature for maximum inactivation has been observed for the milk-spoilage proteases from different *Pseudomonas* species [6,18]. On the other hand, less AprX activity was lost in milk than in PBS ($p < 0.01$), indicating that milk proteins reduced AprX inactivation by autoproteolysis, which may be due to the milk proteins acting as a competitive substrate for AprX proteolysis [11,42], and due to the steric hindrance by the formation of reversible protease–caseinate complexes [12]. A small shift in the maximal inactivation temperature was observed in milk samples (60 °C) compared to that in PBS samples (58 °C). The shift is suggested to have been caused by the protective effect of milk, i.e., milk constituents stabilize the native form of AprX, leading to an increase in the denaturation temperature [5].

When looking at the parent proteins in the peptidomic data, the non-incubated control sample already contained a wide range of peptides, especially originating from β-casein and $\alpha_{s1}$-casein (Figure 2). Not only were a large number of peptides already found in control milk samples, but they also covered a large part of the protein sequences, especially for the caseins (Figure 3). Similar profiles of peptides have also been reported previously in non-inoculated UHT milk [43] and pasteurized milk [27]. These peptides could be naturally occurring in bovine milk by the actions of native proteases such as plasmin (Figure S1) [44,45], they could have resulted from the cleavage of peptide bonds during severe UHT heating, or they could have been formed during storage [46,47].

Compared to the non-incubated control samples, a large increase in the summed intensities of peptides for all abundant milk proteins could be seen in all incubated samples (Figure 2). This increase indicates extensive hydrolysis of the milk proteins by AprX, both with and without LTI treatment. Next to that, when determining the preference of the enzyme towards the different abundant milk proteins, the results differed depending on whether a qualitative or quantitative approach was chosen. After incubation, β-casein and $\alpha_{s1}$-casein were represented by the largest number of peptides, whereas the total intensity increased the most for the peptides from κ-casein and β-casein. This indicates that κ-casein and β-casein were the most sensitive towards hydrolysis by AprX. These results are in accordance with the consensus that both κ-casein and β-casein are preferential substrates for extracellular bacterial proteases [11,31,48–50]. This preferential hydrolysis of β-casein by AprX may be because of its high concentration in the aqueous phase and the particular free position in casein micelles [51], with the cleavage of κ-casein probably being caused by its position on the surface of the casein micelle. The α-caseins have also been reported to be substrates for AprX, although to a lesser extent than κ- and β-caseins [5]. However, no increase was found in the intensity of $\alpha_{s1}$- and $\alpha_{s2}$-casein-derived peptides compared to control samples. This indicates that $\alpha_s$-caseins are relatively insensitive towards AprX hydrolysis.

Although the increase in κ-casein peptide intensities was highest among the different milk proteins, the number of different peptides formed was rather low. The low number of identified peptides may be because the hydrolysis of κ-casein was focused on the peptide bonds Ger104-Phe105 and Phe105-Met106 (Figure 3(B2)), which is similar to previous studies describing the AprX-induced proteolysis of κ-casein [9,28,30,52]. Compared with chymosin in cheese making, AprX does not only cleave the peptide bond Phe105-Met106 of κ-casein, but it also non-specifically cleaves around the region of soluble hydrophilic glycomacropeptide (Figure 3(B2)), which, as with chymosin hydrolysis, leads to casein micelle destabilization.

For $\alpha_{s1}$- and $\alpha_{s2}$-caseins, only the phosphorylated regions were not well covered (Figure 3C,D), which may be due to these peptides typically being less well ionized, making it more difficult to identify them by LC-MS/MS [53].

Next to the caseins, β-lactoglobulin was also found to be hydrolyzed by AprX (Figure 2). Whey proteins have been considered so far to be relatively insensitive to the action of psychotropic bacterial proteases [8,54,55]. This resistance to proteolysis is thought to be caused by its compactly folded structure [56], which may explain the low number and intensity of β-lactoglobulin-derived peptides in FM (Figure 3(E1)). Nevertheless, based on our results, β-lactoglobulin was also readily hydrolyzed by AprX, as also observed on the electrophoretic gel in our previous study [5]. It is noteworthy that after incubation with AprX, the peptide number and intensity increased markedly, but the peptides originated from the same area of the protein sequence (Figure 3(E2)). This implies that AprX hydrolyzed β-lactoglobulin by cleaving the same areas as the native milk proteases.

The peptidomic profile of all milk proteins after the action of AprX has also been mapped before [11,28,43]. However, the number of detected peptides in the present study was significantly higher than in these studies, which may be attributed to the different analytical approaches, making a direct comparison between these studies difficult.

With regard to the enzyme specificity, the relative frequencies of amino acids at the P1 and P1′ positions were determined (Figure 4), which showed that Asp, Glu, and Cys in the P1 and P1′ positions, Ile in the P1 position, and Pro in the P1′ position were less cleaved. Matéos et al. [11] showed that when purified caseins were incubated with the *Pseudomonas* LBSA1 extracellular protease for 24 h, peptide bonds around basic amino acid residues and Phe were preferentially cleaved, while at the same time, peptide bonds around cysteine and proline were poorly cleaved, which is in accordance with our findings. However, the preference and disfavor of AprX for other amino acid residues as reported in that study were not replicated in our results. These different findings may be explained by differences in the enzyme selectivity at different stages of hydrolysis [57], as well as the high AprX concentration used and differences in the analytical methods used.

Finally, when comparing samples with and without LTI treatment, the results did not show generally increased or decreased peptide levels (Figure 5). For skim milk, a few more peptides were higher in the sample with LTI, whereas for full-fat milk, this was the opposite. In addition, when it comes to the predicted bitterness, no consistent results were found (Figure 6). The minor differences in the peptide profile for samples treated with LTI versus without indicated that the cleavage specificity did not change due to the LTI treatment. Overall, for both peptide release as well as potential bitterness, no consistent conclusion on a possible advantage of LTI could thus be drawn.

## 5. Conclusions

The present study showed that the protease AprX produced by *Pseudomonas* can hydrolyze proteins in UHT milk, particularly κ-casein and β-casein. The identification of the peptides showed a high sequence coverage, and an analysis of the cleavage sites revealed that AprX does not have a strong specificity for specific amino acids. An LTI at 60 °C for 15 min partially inactivated the AprX activity in UHT milk. However, this process did not significantly reduce the subsequent proteolysis of milk proteins, especially in full-fat UHT milk. This indicates that after LTI, besides AprX itself, mainly the milk proteins were hydrolyzed. Therefore, the feasibility of using a low-temperature heat treatment as a means of inactivating AprX may be limited in UHT milk.

**Supplementary Materials:** The following supporting information can be downloaded at: https://www.mdpi.com/article/10.3390/dairy4010011/s1, Figure S1: Scatter plot of the total sites cleaved by an enzyme, at termini (*x*-axis) and log odds ratio (*y*-axis). Enzymes that are the most likely to be active in milk are those with a combined high number of total cleavages and a high odds ratio, represented by the topmost right corner. For the enzymes that do not have expression in milk, data indicate the presence of other enzymes with similar activity. Figure S2: Peptidomic profile from β-casein (A), κ-casein (B), $\alpha_{s1}$-casein (C), $\alpha_{s2}$-casein (D), and β-lactoglobulin (E) in SM, SM + AprX,

SM + AprX + LTI, and FM + AprX + LTI, respectively, analyzed by a nano-LC/LTQ and Orbitrap. MS/MS: Peptides with different intensities were distinguished by color and width. The intensity shown in the figure is the average value of the results in triplicate. Table S1: Details of parent proteins from which peptides were derived. The highlighted rows indicate that peptides from this parent protein were detected to have significantly higher intensities in the AprX-hydrolyzed samples than the blank UHT milk samples. Table S2: Significantly different peptides in samples with and without LTI treatment.

**Author Contributions:** Conceptualization, K.H. and C.Z.; methodology, C.Z. and S.B.; validation, S.B.; formal analysis, C.Z. and S.B.; investigation, C.Z. and S.B.; resources, K.H.; data curation, S.B.; writing—original draft preparation, C.Z.; writing—review and editing, S.B., L.Z., E.B. and K.H.; visualization, C.Z.; supervision, E.B. and K.H.; project administration, K.H. and E.B.; funding acquisition, K.H. All authors have read and agreed to the published version of the manuscript.

**Funding:** This research was funded by the Sino Dutch Dairy Development Center, which aims to improve dairy production, safety, and quality levels throughout the entire dairy chain in China, and the China Postdoctoral Science Foundation (2020M681210).

**Data Availability Statement:** All data used for the manuscript have been made available as Supplementary Data Files.

**Conflicts of Interest:** The authors declare no conflict of interest. The funders had no role in the design of the study; in the collection, analyses, or interpretation of data; in the writing of the manuscript; or in the decision to publish the results.

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
