# Peer review of "The Impact of Low-Temperature Inactivation of Protease AprX from Pseudomonas on Its Proteolytic Capacity and Specificity: A Peptidomic Study"

_2624-862X, doi:10.3390/dairy4010011_

Round 1

Reviewer 1 Report

The manuscript reports a study aimed at characterizing the proteolytic activity of the AprX, i.e. the protease typically produced by Pseudomonas and responsible of UHT milk destabilization. Commercial UHT milk, either skimmed or full-fat, was intentionally added with the crude enzyme (0.2 mg/mL) and the related activity was evaluated after milk incubation at different temperatures for 15 min. The effectiveness of a low-temperature inactivation (LTI) of AprX by heating milk at 60 °C for 15 min was also tested. The proteolytic activity (both qualitative and quantitative) on the major milk proteins was evaluated by a peptidomic approach and data were statistically evaluated.

Based on the title, the topic of the manuscript looks to be of great interest to the dairy industry worldwide, since AprX inactivation at plant milk processing is still an unsolved issue. However, the study then is mainly addressed at identifying the specific sites of AprX action on individual milk proteins, including preferentially cleaved amino acid residues. This is an interesting analytical exercise, contributing significant knowledge to the scientific community but scarcely responding to reader expectation. The scientific literature dealing with AprX activity is huge and thousands of derived peptides have been identified. It is confirmed that k-casein and b-casein are preferential substrates for the studied protease in milk. However, the applicative impact of these studies is not so evident.

Despite this criticism, the manuscript has undoubted scientific merits. The experimental design of the study is well defined and accurately described. The adopted analytical techniques are sound, thus giving guarantee of result reliability.

I recommend the authors to change the title to make it more reflecting the actual content of the manuscript. In the present form, title is misleading. The inadequate inactivation of AprX by LTI is one among various outputs of the study, and it was already evidenced in Figure 1. In respect of LTI, the focus was mainly on the changes in cleavage specificity induced by this process, and thus on the peptide patterns in LTI compared with non-LTI samples. The reasons behind the observed changes are not investigated and a discussion in the light of existing literature almost lacks.  

Another concern is about the cited references. Around 30% of these are more than 20 years old, some more than 50.  Besides the aim of giving evidence of the dated interest for this topic, I recommend a more careful consideration of the recent literature. This would help to introduce more convincing comments in the discussion.

I noticed that ref. [11] and [39] look to be the same.

Overall, the manuscript is well written and its scientific merit is satisfactory. The recommended changes will help fully targeting the actual topics of the study.

Author Response

I recommend the authors to change the title to make it more reflecting the actual content of the manuscript. In the present form, title is misleading. The inadequate inactivation of AprX by LTI is one among various outputs of the study, and it was already evidenced in Figure 1. In respect of LTI, the focus was mainly on the changes in cleavage specificity induced by this process, and thus on the peptide patterns in LTI compared with non-LTI samples. The reasons behind the observed changes are not investigated and a discussion in the light of existing literature almost lacks.  

AU: we agree that the title may indeed may confusing to the readers. We have therefore changed the title to:

“The impact of low temperature inactivation of protease AprX from Pseudomonas on its proteolytic capacity and specificity: a peptidomics study”

Another concern is about the cited references. Around 30% of these are more than 20 years old, some more than 50.  Besides the aim of giving evidence of the dated interest for this topic, I recommend a more careful consideration of the recent literature. This would help to introduce more convincing comments in the discussion.

I noticed that ref. [11] and [39] look to be the same.

AU: Several recent references (ref 7, 26, 27, 31 & 32) have been added and the mistake has been fixed. However, as this field of research has been active studied before 2000, with some important studies especially about LTI from this time, we decided to keep also the older references in the manuscript.

Reviewer 2 Report

REVIEW

for the journal Dairy (ISSN 2624-862X)

Article “Low temperature inactivation of protease AprX from Pseudomonas may not be feasible in UHT milk: a peptidomic study

Manuscript ID: dairy-2146070

Authors:  Chunyue Zhang, Sjef Boeren, Liming Zhao, Etske Bijl  and Kasper Hettinga

This study aims to increase the existing, but insufficient knowledge of AprX hydrolysis patterns on milk proteins by using a quantitative peptidomics perspective and to determine the feasibility of using an LTI treatment as a means of inactivating AprX in both full fat and skim UHT milk, by comparing the peptidome of AprX-hydrolyzed UHT milk samples with and without a LTI treatment. This knowledge is important for milk producers and consumers.

The methodology of the study, validity of the results and conclusions leave no doubt, but I would like to make a few comments that could improve the quality of this interesting article.

1) In my opinion, section "2.7 Data Analysis" should be more detailed and consistent. Authors should describe the study samples, compared groups, taking into account the design of the study. I also recommend providing a summary of what software was used for data processing and various statistical tests.

2) Lines 308-309. "To identify the potential enzymes responsible for cleaving the proteins, a web-based software EnzymePredictor was used". This information should be in the methodology section of the article.

3) Lines 391-393 "Bitterness due to formation of hydrophobic peptides is frequently encountered in UHT milk [35, 36]. In our study, in total 877 peptides were identified to have a Q value above 1400 cal/mol, which is an evaluation criteria of a bitter peptide [36] ".

In my opinion, this information should be in the literature review or discussion section.

4) Line 545. "2015" should be in bold.

5) Bibliographic description of the literary source No. 36 - incomplete.

6) The same for source 40.

7)  Line 641: “Page 934”. The bibliographic description needs to be corrected.

8) The authors should clarify the descriptions of 47, 49 literary sources.

 The article is interesting, but the adjustments mentioned are recommended.

Sincerely, reviewer.

Author Response

1) In my opinion, section "2.7 Data Analysis" should be more detailed and consistent. Authors should describe the study samples, compared groups, taking into account the design of the study. I also recommend providing a summary of what software was used for data processing and various statistical tests.

AU: Details on both the number of samples as well as the software tools has been added to section 2.7.

2) Lines 308-309. "To identify the potential enzymes responsible for cleaving the proteins, a web-based software EnzymePredictor was used". This information should be in the methodology section of the article.

AU: Agreed, this has been removed from results and text was rephrased accordingly.

3) Lines 391-393 "Bitterness due to formation of hydrophobic peptides is frequently encountered in UHT milk [35, 36]. In our study, in total 877 peptides were identified to have a Q value above 1400 cal/mol, which is an evaluation criteria of a bitter peptide [36] ".

In my opinion, this information should be in the literature review or discussion section.

AU: This was moved to the introduction.

4) Line 545. "2015" should be in bold.

5) Bibliographic description of the literary source No. 36 - incomplete.

6) The same for source 40.

AU: References have been made correct and complete (also another one that was incomplete). These are references 4, 11, 21 & 44 in the revised manuscript.

7)  Line 641: “Page 934”. The bibliographic description needs to be corrected.

AU: We believe this reference to be correct.

8) The authors should clarify the descriptions of 47, 49 literary sources. The article is interesting, but the adjustments mentioned are recommended.

AU: These references have been made complete for easier looking up (ref. 51&52 now).